# Transgenic *Drosophila* Expressing Active Human LH Receptor in the Gonads Exhibit a Decreased Fecundity: Towards a Platform to Identify New Orally Active Modulators of Gonadotropin Receptor Activity

**DOI:** 10.3390/ph17101267

**Published:** 2024-09-25

**Authors:** Amir Mahamid, David Ben-Menahem

**Affiliations:** Clinical Biochemistry and Pharmacology, Faculty of Health Sciences, Ben-Gurion University of the Negev, Beer-Sheba 8410501, Israel

**Keywords:** transgenic *Drosophila melanogaster* (*Drosophila*), gonadotropins, luteinizing hormone receptor (LHR), drug screen

## Abstract

Background/Objectives: The gonadotropins luteinizing hormone (LH) and follicle-stimulating hormone (FSH) and their receptors are major regulators of reproduction in mammals and are absent in insects. We previously established transgenic *Drosophila* lines expressing a constitutively active human LH receptor variant (LHR^D578Y^) and the wild-type receptor (LHR^wt^; inactive in the absence of an agonist). That study showed that ubiquitously expression of LHR^D578Y^—but not of LHR^wt^—resulted in pupal lethality, and targeted expression in midline cells resulted in thorax/bristles defects. To further study the *Drosophila* model for an in vivo drug screening platform, we investigated here whether expressing LHR^D578Y^ in the fly gonads alters reproduction, as shown in a transgenic mice model. Methods: The receptor was expressed in somatic cells of the gonads using the tissue-specific *traffic jam*-Gal4 driver. Western blot analysis confirmed receptor expression in the ovaries. Results: A fecundity assay indicated that the ectopic expression of LHR^D578Y^ resulted in a decrease in egg laying compared to control flies carrying, but not expressing the transgene (~40% decrease in two independent fly lines, *p* < 0.001). No significant reduction in the number of laid eggs was seen in flies expressing the LHR^WT^ (<10% decrease compared to non-driven flies, *p* > 0.05). The decreased egg laying demonstrates a phenotype of the active receptor in the fly gonads, the prime target organs of the gonadotropins in mammals. We suggest that this versatile *Drosophila* model can be used for the pharmacological search for gonadotropin modulators. Conclusions: This is expected to provide: (a) new mimetic drug candidates (receptor-agonists/signaling-activators) for assisted reproduction treatment, (b) blockers for potential fertility regulation, and (c) leads relevant for the purpose of managing extra gonadotropic reported activities.

## 1. Introduction

The gonadotropins luteinizing hormone (LH), human choriogonadotropin (hCG), and follicle-stimulating hormone (FSH) are glycoproteins secreted from pituitary gonadotropes in response to gonadotropin-releasing hormone (GnRH) released from the hypothalamus [1,2]. In the gonads, LH and FSH control the production of steroid hormones (steroidogenesis) critical to follicular maturation and ovulation as well as spermatogenesis [3,4,5,6]. HCG is expressed in the placenta of primates and equids, and it interacts with the LH receptor (LHR) to maintain early pregnancy in primates [7,8,9,10]. Each hormone is a non-covalent heterodimer composed of a common a subunit and a hormone-unique β subunit that confers receptor specificity to the heterodimer [9].

Along the hypothalamic-pituitary-gonad axis both GnRH and the gonadotropins interact with specific G-protein coupled receptors (GPCRs) [2,7,11,12]. Gain- and loss-of-function mutations in the genes for the gonadotropin receptors have been extensively described and can cause severe pathological effects including infertility and male precocious puberty (reviewed in [2,7,11]). The gonadotropins are extensively used in assisted reproduction protocols, and their administration requires injections. Till now, no orally active gonadotropin modulators are available for clinical use. This absence, despite an eminent need, is due in part to the complexity and high costs of drug candidate screening methods based exclusively on conventional mammalian-derived models.

***Drosophila melanogaster as a model organism to study human physiology and pathological states*:** Drosophila models traditionally provided excellent opportunities to identify novel signaling mediators and also exploited to identify drug candidates [13,14,15]. The remarkable validity of *Drosophila* for human health and disease research, including the ultimate goal of drug development was proven by a series of incisive studies in which a fly model for multiple endocrine neoplasia type 2 (MEN2) was developed by Cagan and his colleagues [16,17]. The investigators expressed mutated forms of *Drosophila* Ret tyrosine kinase receptor in the developing fly eye, carrying dominant mutations found in patients endocrine organs, including the thyroid (an endocrine gland absent in *Drosophila*), and analyzed the signal transduction pathways, as well as identified genes that contribute to the oncogenic Ret-mediated growth phenotypes. Eventually, a drug screening aimed at identifying edible compounds that rescue the Ret-induced rough-eye-phenotype, which is tractable in the adult fly (disruption of the ommatidia structure), identified ZD65474/Vandetanib as a potential therapeutic compound, taken orally [17]. This work led to clinical trials and in 2011 the tyrosine kinase inhibitor Vandetanib (Caprelsa, AstraZeneca, Cambridge, UK) was available to patients, at the time the only FDA-approved medication for advanced medullary thyroid cancer.

***Drosophila gonadotropin model*:** Although the gonadotropins are absent in insects, substantial information supports the existence in *Drosophila* of structurally and evolutionary-related hormones and receptors and autocrine/paracrine loops that modulate gonadotropin activity in mammals [18,19,20,21,22,23,24,25,26,27,28,29,30,31,32,33,34,35]. These findings support the notion of the conservation of endocrine-signaling mediators in flies and vertebrates. In collaboration with Dr. Ross Cagan (then at Mount Sinai Medical School, New York, NY, USA) we previously observed that ectopic expression of the constitutive active LHR variant, *LHR^D578Y^* variant (Figure 1), in *Drosophila,* led to multiple phenotypes [36]. Pushing the expression of the active receptor with Tubulin and Actin drivers in all animals showed a strong drop in viability: most died as pupae with <10% surviving to adulthood. In addition, the use of the midline glia driver *slit-GAL4* led to significant morphological changes in the thorax in *slit > LHR^D578Y^* (penetrance > 95%) adults including abnormal bristle patterning, a sensitive phenotypic readout [36]. In contrast, over-expression of the LHR^WT^ did not result in a detectable phenotype, indicating low basal activity of the protein in the fly [36]. Importantly, these defects depended on the activity of canonical downstream gonadotropin signaling (PKA/CREB pathway; [36]. Taken together, these data indicate that the ectopically expressed LHR can be active, and crucial aspects of gonadotropin signaling are conserved in *Drosophila*.

Here, we examined whether expressing the human LHR in the fly gonads affects reproduction. Our purpose was to further study the *Drosophila* gonadotropin model by assessing the activity of the receptor in the gonads for the potential application of this platform for pharmacological analysis of the LHR modulators and signaling components (agonists/activators and antagonists/blockers) aimed at drug candidate screening.

## 2. Results

### 2.1. Expression of the Human LHR in the Fly Ovary

The somatic cells in the gonads of mammals are the main site of the gonadotropin receptor expression. To overexpress the wild-type or point-mutated LHR variant in the flies, we used the two components Gal4-UAS system [37]. First, we examined the expression of the LHR in the fly ovary using Western blot analysis. To drive the expression of the human LHR in the gonads, we crossed *TrafficJam-GAL4* flies (*TJ*-*GAL4*) to *UAS-LHR* flies (WT or D578Y), and dissected the ovaries of the progenies (Figure 1 and Figure 1). In females, *TJ*-*GAL4* drives the expression of the UAS-transgene predominantly in the egg chamber (the equivalent of the follicle in mammals), which surrounds, contacts, and communicates with the oocyte [38,39]. Expressing (driven) and non-expressing (non-driven) offsprings can easily be scored, based on the wing appearance. While the wings of the driven flies exhibit a normal morphology (denoted as Non-CyO), the wings of the control non-expressing progeny of the same cross are curled (CyO), because of a dominant mutation that results in the noticed phenotype of a curved shape.

The HA antibody recognized the tagged receptor in the two examined LHR^D578Y^ independent lines (#4 and #15) and the LHR^WT^ driven flies (Non-CyO flies; NCy; Figure 2, lanes 1, 3, and 5). The receptor appeared as few bands, similar to the appearance when ubiquitously expressed in *tub > LHR^D578Y^* pupae (denoted as PNT; Figure 2, lane 7). This is in accordance with our previous observation when ubiquitously expressed with this driver [36]. It is generally accepted that the multi-band LHR appearance is related to receptor maturation; the immature forms appear with a lower molecular mass than the mature form in electrophoretic analysis (for example see [36] and references therein). In contrast, no specific bands appeared in ovaries dissected from non-driven females from the same *TJ*-*GAL4* crosses (CyO flies; Cy; Figure 2, lanes 2, 4, and 6).

As additional controls, we examined the protein expression in the ovary of non-CyO offsprings of *TJ*-*GAL4* mated to *UAS-eGFP* (an unrelated protein) and *W^1118^* offsprings (matched the genetic background of the transgenic flies used in the study). As expected, the receptor was not detected in the ovaries of these pushed flies (NCy; Figure 2 lanes 8 and 9). Anti-alpha tubulin and calreticulin antibodies were used to examine the presence of these housekeeping proteins, for estimating protein loading in all the lanes.

### 2.2. Fecundity of LHR Flies

In the gonads, the somatic and germline cells communicate, and this interaction is important for the development of the egg and sperm of the fly. To examine the effect of the LHR on the fecundity, we used *TJ*-*GAL4* to push the expression. We collected unmated offspring and crossed single pairs of driven and non-driven flies. The expression of the active receptor variant (non-CyO flies) resulted in a 40% in egg laying compared to control CyO flies, which carry -but do not express- the receptor transgene. This was evident in the two independent LHR^D578Y^ fly lines (#4 and #15, *p* < 0.001; Figure 3). In contrast, LHR^WT^ which in mammals is inactive in the absence of a ligand, did not have a significant effect, and expressing the receptor resulted in less than a 10% decrease in the number of eggs laid during the three days of the experiment compared to the non-expressing flies (*p* = 0.13, Figure 3). In the case of the mated eGFP and W^1118^ flies, the number of laid eggs by the CyO and non-CyO progenies was similar (eGFP: ~5% reduction, *p* = 0.69; W^1118^: 1% increase, *p* = 0.97; Figure 3). The decreased fecundity of the driven LHR^D578Y^ flies demonstrates a phenotype of the active human gonadotropin receptor when expressed in the *Drosophila* gonads.

To further examine whether the observed egg-laying decrease is related to the activity of the LHR, we also assessed the fecundity of flies expressing the WT receptor together with a ligand that is known to activate the LHR. In a previous study, these flies (CGβα;LHR^WT^) displayed similar phenotypes to LHR^D578Y^ lines when the expression was driven either ubiquitously (lethality) or predominantly in midline-glia cells (multiple thorax defects) [36]. We used the agonist alone (CGβα) and LHR^WT^ as control flies in this independent set of fecundity experiments (Figure 4). The driven WT receptor together with the ligand flies showed a decrease of some 20% (*p* = 0.013) in the number of laid eggs compared to the control non-driven CGβα;LHR^WT^ flies (CyO). A statistically insignificant slight reduction in the fecundity of the ligand-expressing flies was observed compared to corresponding the non-expressing flies (CGβα; 7% difference, *p* = 0.43) (Figure 4). As above in these experiments also the number of eggs laid was similar in the case of expressing and non-expressing LHR^WT^ crosses (about 5% difference; *p* = 0.52).

### 2.3. Gender Analysis of the Gonad Phenotype

In humans, the known phenotype resulting from the presence of the point mutation at position 578 of the LHR is evident in boys but not in girls—male-limited precocious puberty [40,41,42]. In transgenic mice models of the corresponding mutation in the mouse receptor (KiLHR^D582G^), multiple reproductive abnormalities, including precocious puberty, were noticed in both males and females [43,44]. We hence examined whether the decreased fecundity seen in the flies expressing the point mutated LHR variant is connected to the male, female, or flies from both genders. For this purpose, we mated expressing and non-expressing LHR females with W^1118^ males (Figure 5).

In females, *TJ-Gal4* predominantly drives expression in somatic support cells of the ovary at all developmental stages [38,39]. These cells are equivalent to Granulosa and Theca cells in mammals; the major expression site of the gonadotropin receptors in the ovary. Crossing W^1118^ males to LHR^WT^-expressing females resulted in a 12% decrease in egg laying—statistically insignificant (*p* = 0.25)—compared to the case when the control females that carried, but did not express the LHR (Non-CyO vs. CyO; Figure 5A). In contrast, when the mating was of W^1118^ males and LHR^D578Y^ expressing females, a large decrease in some 35–38% in egg laying compared to the corresponding control CyO crosses was seen (Figure 5A). This was evident in the two examined independent LHR^D578Y^ lines (#4 and #15; 35–38% reduction, *p* < 0.001 in both cases; Figure 5A).

In the reciprocal set of crosses, no significant changes in the number of laid eggs by the W^1118^ females mated with driven compared to non-driven LHR males were observed (Figure 5B). This was seen in the case of WT as well as D578Y LHR males (a 12% difference at the most; *p*= 0.52 (LHR^WT^), *p* = 0.55 (LHR^D578Y−4^) and *p* = 0.48 (LHR^D578Y−15^)) (Figure 5B). Together, these results suggest that the decreased fecundity is primarily due to the female rather than male LHR^D578Y^ flies.

## 3. Discussion

In the current study, we were interested in studying the effect of gonadotropin receptor activity on fly fecundity when expressed in the somatic follicle cells that contact germ cells in the gonads of *Drosophila*. Our study showed that overexpressing the active LHR variant, but not the WT receptor, resulted in a significant reduction in the number of laid eggs, compared to flies from the same cross that do not express the receptor. In addition, the data suggests that this compromise is related to a female -rather than a male- factor/s, which is currently unknown and implies a potential dimorphic reproduction phenotype.

The decreased fecundity associated with the overexpression of the active LHR is in agreement with the view that overstimulation of gonadotropin receptors has negative consequences on reproduction in mammals. This phenomenon was illustrated for both LH and FSH receptors and was reported in various mammalian models. For instance, transgenic mice overexpressing a long-acting LH analog [45] as well as the above-mentioned mice expressing a constitutive active LHR variant showed multiple defects in the gonads including compromised fertility [43,44]. It was shown previously that sustained high gonadotropin/receptor levels affect oocyte maturation and are not good for fertilization (for example see, [46,47,48]). For instance, high LHR mRNA levels in human granulosa cells have been reported to correlate with decreased fertilization in assisted reproduction protocols [49]. Based on a detailed study of female rats, Orisaka and colleagues reported that sustained LH stimulation dysregulates follicular growth, and this involves a decrease in FSH-induced folliclogenesis [50]. Despite obvious differences in the reproductive physiology of humans and flies, some of the basic aspects of follicle development and ovulation are conserved [51,52,53,54,55,56,57]. The newly identified egg-laying phenotype in our study and the powerful genetic tools available in *Drosophila* may assist in understanding potential causes for the decreased fecundity in the fly, and potentially identify novel factors involved in the fine-tuning of ovulation in mammals.

Fertility regulation remains a major health challenge in significant parts of the world. Despite these efforts, no gonadotropin mimetics or antagonists are available for clinical use. The lack of analogs is due, at least in part, to the lack of understanding of the physiology of reproduction and to the extremely high costs of conventional drug discovery techniques. Incorporating the gonadotropin fly platform in the fertility drug development process is expected to improve the efficiency and accelerate the search for edible LH receptor modulators in the context of all animals. This can be beneficial in early drug discovery and development stages, and potential hits can be further examined in relevant mammalian-based models.

In accordance with this view, Sun and his colleagues used recently an ex vivo assay of Drosophila follicle rupture to screen a library of FDA-approved drugs as potential ovulation blockers. The screen was basically based on testing candidates against the activity of biogenic amines which are major ovulation regulators in the fly and identified several compounds that inhibited ex-vivo follicular rupture [58]. Furthermore, some of the drugs showed similar effects on cultured mouse follicles and one compound blocked ovulation in vivo [58]. The identified hits have various indications in humans including for CNS disorders (e.g., the antidepressants amitriptyline and niclosamide and the antipsychotic drug chlorpromazine). While multiple additional central effects of the identified candidates are expected to be associated with the potential anti-ovulation activity in mammals, that study showed the potential of *Drosophila* in the search for new contraceptive candidates.

The combination of reproducible phenotypes observed in the gonadotropin flies by two independent approaches (LHR^D578Y^ and CGβα;LHR^WT^) and the high penetrance are key for identifying specific LH/CG inhibitors. For this goal, fly LHR lines in the active form lines are useful for a stringent search of anti-gonadotropin compounds based on the ability to rescue various phenotypes. A phenotypic screen based on the ability of a compound to rescue thorax deformation when expressed in midline cells, reduce the lethality when ubiquitously pushed, and overcome the decreased fecundity when driven in the ovary. The robust genetic and cell-biology tools available in *Drosophila* on the one end together with the short generation time and the ability to examine a large number of flies on the other end are advantageous for early stages of economical and accelerated drug screen. These, together with specific and easily tracked LHR-related phenotypes -depending on the expression site- provide a unique cost-effective combination that favors using this fly model to identify potential edible modulators of the receptor in vivo. Such hits may be further examined for their possible ability to control fertility. In addition, identified LHR-blocking candidates may be examined for potential relevance for health issues such as Alzheimer’s disease and certain malignancies (e.g., prostate cancer), where a non-classical involvement of LH/CG was reported [59,60,61,62,63,64,65,66,67,68]. On the other hand, the LHR-acquired phenotypes provide a strategy that is likely to be advantageous in searching orally active gonadotropin mimetics. To identify potential gonadotropin-receptor/signaling activators flies expressing the quiescent LHR can be used. Potential LH mimetics are anticipated to phenocopy the outcome of the active receptor when expressed with ubiquitous and tissue-specific drivers and may be considered as ovulation inducers when needed.

In conclusion, the reproduction phenotype further validates our invertebrate model and provides motivation for gonadotropin studies using the transgenic *Drosophila*, as related to the gonads and possibly extragonadotropic LH activities reported in additional tissues. The multiple and easily tractable specific phenotypes observed in our studies combined with the high signal-to-noise ratio, favor exploiting the fly model as a versatile in vivo approach to identifying new opposers and inducers of LH activity.

## 4. Materials and Methods

### 4.1. Fly Strains and Crosses

The generation of UAS-LHR^D578Y^ (two independent lines, on the 2nd and 3rd chromosomes) and UAS-LHR^WT^ transgenic *Drosophila melanogaster* flies was previously described [36]. As a second strategy for receptor activation we generated WT receptor (LHR^WT^) together with a receptor agonist (UAS-CGβα; a single-chain (yoked) hCG analog [69,70]) flies through genetic crosses (denoted as CGβα;LHR^WT^ [36]). W^1118^ was used as wild-type flies and UAS-eGFP flies (Bloomington #5431) were used as a generic protein expression control flies. Flies were reared on a standard fly food containing cornmeal, yeast, glucose and sucrose, maintained and crossed in a 25 °C incubator (25 ± 2 °C). For all experiments, we used virgin females (4–5 days old) and males as indicated. To drive expression in somatic cells of the gonads at all developmental stages we used *TrafficJam-GAL4* flies [38,39] (denoted as *TJ*-*GAL4*, the enhancer *TJ* allele is over a CyO balancer chromosome; a kind gift from Lilach Gilboa’s lab, formerly at the Weizmann Institute of Science, Rehovot, Israel (Figure 1).

### 4.2. Western Blot Analysis of Receptor Expression in the Ovaries and Pupae

*TJ*-*GAL4* females were crossed to males of UAS-LHR lines carrying the point-mutated or WT receptor variants to express the transgenes in the somatic cells of the gonads, and to UAS-eGFP and W^1118^ males as control flies. Non-driven (curly wings phenotype) and driven (exhibit a normal wing phenotype) female progeny were collected and aged for 4 days at 25 °C (dry yeast were added to the fly food for the last 24 h). On the fourth day, the ovaries were dissected and placed in a gel loading solution (Bio Rad, Hercules, CA, USA) supplemented with β-Mercaptoethanol, in a ratio of two ovaries to 10 μL of sample buffer. The ovaries were ground with a small plastic pestle and heated at 95 °C for 4 min, cooled in ice for 5 min, and stored at −80 °C until analyzed. We also examined LHR^D578Y^ expression when ubiquitously driven with *Tubulin*-*GAL4*. Pupae progeny that express the receptor have been collected and processed for Western blot analysis as previously described [36]. Protein samples were analyzed in 10% SDS-PAGE and transferred to the nitrocellulose membrane. To detect the LHR, it contains an HA tag at the amino terminus; Figure 1. This tag is known not to change the LHR activity [71,72]—in the samples, the membranes were blocked and probed with an anti-HA High-Affinity monoclonal antibody conjugated to HRP (3F10; 1:750; Roche). Following this step, mouse anti-α-tubulin (1:15,000; Sigma, STL, MO, USA) and polyclonal anti-calreticulin (1:1000; Thermo Fisher Scientific, Waltham, MA, USA) antibodies were used in the same membranes for examining loading levels (loading controls). HRP-conjugated anti-rabbit and mouse antibodies were purchased from Jackson Immuno Research Laboratories (1:15,000; west Grove, PA, USA), and the blots were visualized with ECL (EZ-ECL, Biological Industries, Beit-Haemek, Israel). Each analysis was repeated at least three times with similar results. A representative blot is depicted, and the intensity of the probed proteins was quantified using the Image Studio™ Lite software. For each of the antibodies, the signal for LHR^D578Y−4^ Non-CyO (lane 1) was set as 1, and the relative protein level was marked on the lanes.

### 4.3. Fecundity Test

*TJ*-*GAL4* females were crossed to males of the designated genotype, and virgin female and male progenies were collected, kept separately, and aged for 4–5 days. For the experiment, the flies were immobilized on ice (“anesthetized”), and single pairs of males and females of the same genotype were placed in plastic vials with food in a 25 °C incubator for three days. Every 24 h, the flies were flipped to a new vial with fresh food. The number of laid eggs in the empty vial was counted under a stereomicroscope and summed along a three-day period for each female. In a complementary experiment, LHR expressing or non-expressing control females of the designated genotype were individually mated to W^1118^ males as above. In addition, the number of laid eggs was also examined in single crosses of LHR males and W^1118^ females. Three to six independent replicates were performed for each genotype (N = 3–6), and the total number of paired flies examined (n) is indicated in the figure legends.

### 4.4. Statistics

The average number of eggs laid per female during the 3 days of the experiment and the standard error of the mean (SEM) were calculated and plotted. The Student’s *t*-test was used to examine the difference between expressing to non-expressing flies for each genotype. *p* < 0.05 was considered significant; * *p* < 0.05, *** *p* < 0.001.

## Data Availability

Data is contained within the article.

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
