# Peer review of "Transgenic Drosophila Expressing Active Human LH Receptor in the Gonads Exhibit a Decreased Fecundity: Towards a Platform to Identify New Orally Active Modulators of Gonadotropin Receptor Activity"

_pharmaceuticals, 2024, doi:10.3390/ph17101267_

Round 1
Reviewer 1 Report
Comments and Suggestions for Authors
This manuscript utilizes the previous established transgenic Drosophilia lines to show that overexpressing the active LHR variant, rather than the WT receptor, leads to a significant reduction in the number of laid eggs. The observed impact on the reproductive function in flys in this study underscores the potential for gonadotropin studies using the transgenic Drosophila, particularly in exploring the pharmacological search of gonadotropin modulators. The writing of the manuscript is generally clear, but there are still certain aspects need to be improved.
1. In the title, the authors emphasized that the research results supported the development of a platform to accelerate the identification of new orally active modulators of gonadotropin receptor activity. However, within the text, there is a lack of sufficient evidence supporting the applications related to the development of orally active agents. You need to supplement some experiments and clear discussions to prove that.
2. In the first sentence of abstract, the full forms corresponding to the abbreviations are incorrect. In the mammalian physiology condition, LH and FSH stand for follicle-stimulating hormone and luteinizing hormone, separately. These acronyms are used for the hormones produced internally, not for exogenous recombinant hormones, such as lutropin or follitropin.
3. For the last sentence in an abstract, it is recommended to add a colon after "provide" and a comma before "and (c)."
4. There are still some spaces, punctuation, font size and unit writing errors in the main text that need to be carefully reviewed and corrected. For example, the writing of temperature units, the international unit for hours is 'h' and not 'hr', etc.
Author Response
Comment 1: We appreciate the reviewer comment. We added text in the Discussion section to emphasize the utility of our model. We are currently initiating experiments towards this and plan to publish our data in the future.
Comment 2: We made the requested changes of the hormone names in the abstract.
Comments 3 and 4: We thank the reviewer, and made the suggested changes in the last sentence of the abstract (a colon mark and a comma; comment 3) and corrected the writing of temperature and international time units for hours (comment 4).
Reviewer 2 Report
Comments and Suggestions for Authors
The manuscript by Mahamid et al. developed a drosophila model as a drug screen platform for human LH receptor. They compared the fecundity of drosophila upon expressing the LHR wildtype as well as the variant version LHRD578Y in follicle cells, meanwhile the authors also compared the phenotype of drosophila with mouse when expressing LHR. While this work is innovative and offers a direction for drug screen in drosophila which is more effective and economical, the manuscript could benefit from more explanations of the results and carefully writing to aid understanding.
Concerns:
11. In Figure 2, there are several bands showed in the western blot experiment, could authors explain why there are few bands on the membrane?
22. About the fecundity test, what is the hatching rate when expressing LHRWT or LHRD578Y? Could authors test this?
3. The whole paragraph “To further examine that egg laying decrease is related…”, can authors checking the writing carefully? There are several mistakes about writings.
4. In Figure 4 figure legend, what does the “n” mean here? Please clarify it.
Comments on the Quality of English LanguageOverall, the English is OK, but the authors should carefully check their writing, there are several typos and mitakes in this manuscript.
Author Response
We appreciate the reviewer general comment and accordingly added text in the discussion to further explain the benefit of the Drosophila model in face of the results.
1st concern: Multiple bands are typical in Western Blot analysis of the LHR. It is generally accepted among experts in the field that this is related to immature and mature receptor forms. In our previous manuscript by Graves et al., MCE, 2015 entitled "The LH/CG receptor activates canonical signaling pathway when expressed in Drosophila" (reference #36 in the current manuscript), we referred to this point and also provided references to the LHR multi-band appearance. As the reviewer recommended, explanatory text is now added in the results section to clarify this point (2.1: expression of the human LHR in the fly ovary).
2nd concern: The reviewer raised an intriguing point but we cannot test it in the current study. The TrafficJam-Gal 4 driver is an heterozygous line that includes the CyO balancer chromosome, and was crossed to homozygous LHR lines. From these crosses, we collected progenies -that are all heterozygous- and mated them for the experiments, as specify in the Material and Methods and Results sections. Hence, the genotypes of the counted eggs are heterogenous, and the enclosing flies from these eggs consist of multiple genotypes. Because of these genetics it is literally impossible to accurately determine in our fecundity test the hatching rates of the expressing LHRWT and LHRD578Y flies. Of note, in our previous study (reference #36) we observed that when driven with various tissue-specific both LHRWT and LHRD578Y expressing were viable, but hatching rate was very low when ubiquitously driven with Tubulin-Gal4 and Actin-Gal4 drivers.
3rd concern: We agree and appreciate the reviewer comment. We checked carefully and corrected the text in the mentioned paragraph (last paragraph in the 2.2 result section named "Fecundity of LHR flies").
4th concern: The reviewer asked what in the figure legends is the meaning of the "n=…". This "n" rerefers to the number tested single-paired flies, and it is now clarified in the last sentence of paragraph entitled "4.3 Fecundity test" in the Materials and methods section.
Reviewer 3 Report
Comments and Suggestions for Authors
1. The title is too long. It is not easy to be quickly understood.
2. The Scheme 1 in Result 2.1 is not helpful to understand the related description "While the wings of the driven flies exhibit a normal morphology (denoted as Non-CyO), the wings of the control non-expressing progeny of the same cross are curled (CyO), because of a dominant muta-tion that results in the noticed phenotype of a curved shape." The reviewer suggests the authors develop a new and easy understanding Scheme to help readers understand their results.
3. The data is limited that can not match the reputation of the journal, in my view.
Author Response
Comment 1: We appreciate the reviewer comment, and shortened the title with the hope that it is better now.
Comment 2: We modified the scheme (Scheme 1 in Results 2.1) and added a new simple illustration of the flies to make it more clear and helpful (Scheme 1B). The legend for the scheme is modified accordingly.
Comment 3: I appreciate the reviewer's last comment but respectfully disagree with it; the data is novel, and strongly support the fly model as a potential platform for fertility drug screen. This is in line with the public interest for new fertility drugs with alternative route of administration that are highly needed.
Round 2
Reviewer 3 Report
Comments and Suggestions for Authors
The authors have replied my concerns one by one and the manuscript quality has been improved greatly. I have no more comments.